# Review

behaviour, ecology, evolution

sour, evolution, taste, acidity, fermentation

**Author for correspondence:**
Robert R. Dunn
e-mail: rrdunn@ncsu.edu

# The evolution of sour taste

Hannah E. R. Frank[1], Katie Amato[3], Michelle Trautwein[4], Paula Maia[3], Emily R. Liman[5], Lauren M. Nichols[2], Kurt Schwenk[6], Paul A. S. Breslin[7,8] and Robert R. Dunn[2,9]

[1]Department of Crop and Soil Sciences and [2]Department of Applied Ecology, North Carolina State University, Raleigh, USA
[3]Department of Anthropology, Northwestern University, Evanston, IL, USA
[4]Entomology Department, Institute for Biodiversity Science and Sustainability, California Academy of Sciences, San Francisco, USA
[5]Department of Biological Sciences, Section of Neurobiology, University of Southern California, Los Angeles, CA, USA
[6]Department of Ecology and Evolutionary Biology, University of Connecticut, Storrs, CT, USA
[7]Department of Nutritional Sciences, Rutgers The State University of New Jersey, New Brunswick, NJ, USA
[8]Monell Chemical Senses Center, Philadelphia, PA, USA
[9]Center for Evolutionary Hologenomics, University of Copenhagen, Copenhagen, Denmark

KS, 0000-0002-0767-3940; RRD, 0000-0002-6030-4837

The evolutionary history of sour taste has been little studied. Through a combination of literature review and trait mapping on the vertebrate phylogenetic tree, we consider the origin of sour taste, potential cases of the loss of sour taste, and those factors that might have favoured changes in the valence of sour taste—from aversive to appealing. We reconstruct sour taste as having evolved in ancient fish. By contrast to other tastes, sour taste does not appear to have been lost in any major vertebrate taxa. For most species, sour taste is aversive. Animals, including humans, that enjoy the sour taste triggered by acidic foods are exceptional. We conclude by considering why sour taste evolved, why it might have persisted as vertebrates made the transition to land and what factors might have favoured the preference for sour-tasting, acidic foods, particularly in hominins, such as humans.

## 1. Introduction

Sour taste is one of five major taste qualities perceived by humans, along with salty, sweet, umami (savoury) and bitter [1]. Nearly all (or perhaps even all) modern human cultures employ microbes in ways that make foods that are not sour become more sour [2], and the preparation of such sour ferments pre-dates the origins of agriculture [3,4]. Human babies are born able to recognize sour tastes (they pucker their lips; [5]). Yet, sour taste remains more mysterious than the other human taste qualities. We do not know why sour taste evolved, nor how it has changed evolutionarily among species and lineages. We do not even know, for example, whether sour taste has evolved just once within vertebrates or multiple times. Most mentions of the adaptive role of vertebrate sour taste are found in the discussion sections of papers focused on the physiology of sour taste [6–9] or briefly in broader reviews [10,11]. The most comprehensive treatment to date is in a conceptual review by one of us on the evolution of human taste [10].

For other sensory systems, genetic analysis of sensory signalling molecules has provided insight into the evolution of that system [12,13] (see below). The same type of analyses have not been possible for sour taste. A receptor for sour taste in vertebrates was recently identified as OTOP1, an unusual type of protein which enables protons ($H^+$ ions) to cross cell membranes [14]. Mice with a genetic inactivation of the gene encoding OTOP1 show deficits in cellular and neural gustatory responses to acids, although they retain behavioural sensitivity (aversiveness) to acids mediated by trigeminal fibres [6,15]. However, since mice lack a strong appetitive component of sour taste, it is unknown whether

OTOP1 contributes to the appetitive phase of acid liking in other species. OTOP1 is conserved across vertebrate species, and OTOP1-like proteins with similar functional properties are found in organisms as evolutionarily distant as humans and vinegar flies ('Drosophilids') [14,16,17]. However, genetic analysis of the gene encoding OTOP1 is unlikely to yield insight into evolutionary pressure on sour taste as it is expressed widely, including in the middle ear. As a result, differences in OTOP1 among species or populations need not necessarily be due to selection related to sour taste, nor have changes in sour taste as their primary effect. For example, some mutations in OTOP1 are associated with vestibular disorders [18].

We know which compounds tend to stimulate sour taste receptors in vertebrates. Sour taste is elicited by both organic acids, including lactic acid, citric acid, malic acid and acetic acid, and inorganic acids, such as hydrochloric acid, nitric acid and sulfuric acid [19]. In the case of strong inorganic acids, the stimulus that triggers sour taste is the proton (a positively charged hydrogen ion). As a result, the sourness of inorganic, fully dissociating acids is directly related to their pH (and hence the availability of protons). For the weak, partially dissociating organic acids, sour taste is related both to the concentration of free protons (and hence pH) as well as to the concentration of protonated organic acids [20]. As a result, organic acids are perceived as more sour than are inorganic acids when tasted at the same acidic pH. For example, citric acid with a pH of 3.0 is perceived to be more acidic than is hydrochloric acid (HCl) at the same pH. At least all of this is true with regard to sour taste, *per se*.

In vertebrates, the sour taste of acids can be either pleasing (preferred in food choice experiments) or displeasing (not preferred) depending upon intensity, context, the species and other factors. In all species so far studied in any detail, the pleasantness of acids is contingent on concentration. For most acid-liking vertebrate species, the attractiveness of oral acids increases with increasing acidity and then decreases beyond some maximal concentration. This function is often described as an 'inverted-U shaped preference–aversion pattern' [21]. It can be divided into two components, a positive affective or liking component (the first rise of the function) and a negative affective or disliking component. The first component is thought to be determined primarily or even exclusively by the sour taste receptor. The second component, on the other hand, is determined by both the taste receptor and, potentially, by receptors associated with nociception or pain. As a result, preferences for acidic stimuli are very likely to be due to sour taste and aversions to acidic stimuli, particularly those with very low pH values, may be due to a mix of taste and irritation [22]. Species can differ in the slopes of the components of the preference–aversion function. They can differ in the breadth or height of the peak of the function (the so-called 'bliss point'). In addition, although this is poorly explored, they may differ in the extent to which preference–aversion functions (see electronic supplementary material, figure S1a) vary among acid types or among individual acids. Evolutionarily, we expect different features of these functions to evolve in different ways based on the roles sour taste plays in their lifestyles.

Our approach in studying the evolution of sour taste has four considerations. First, we compiled a database of vertebrate species for which the ability to detect acidity in food has been assessed. We sought to note those species able to detect acidity, as well as those unable to detect acidity (see

electronic supplementary material, methods and table S2, second column). Second, we then used this database first to code vertebrates based on whether or not they could detect acidic stimuli, and second on the valence of their response to such stimuli (i.e. dislike, like or variable/uncertain; see electronic supplementary material, methods and table S2). In considering these valences, we include data from studies in which species have a preference for acidic foods relative to non-acidic foods in the wild. Third, we mapped the evolution of the ability to detect acidity and the valence of responses to that acidity on the vertebrate phylogeny, with a special focus on primates, including humans. Fourth, in light of this phylogenetic context, we considered the evolution of sour taste. Throughout, our approach is explicitly interdisciplinary. This extends to the composition of our team which includes ecologists (Dunn, Nichols), microbiological ecologists (Frank, Maia), primatologists (Amato), an evolutionary biologist (Trautwein), an expert on tongue evolution (Schwenk), a specialist on taste evolution/perception (Breslin) and an expert on sour taste transduction (Liman).

We found (electronic supplementary material, table S2) that a remarkably small number of species have been studied with regard to their ability to detect acids. Within mammals, many orders have not been considered (figure 1) and we were able to find data for only 33 species (out of roughly 5400 species of mammals on Earth; [23]). Outside of the mammals, records are even more scarce. Data on sour taste perception were available for just six of the roughly 9900 bird species on Earth [24]. In addition, in the vast majority of the cases listed in electronic supplementary material, table S2 what has been documented is the ability of the animal to detect and respond to acidic substances, yet the mechanism by which they have responded is not necessarily well understood. It is likely that in most cases this ability is reliant on acid taste receptors. However, as we have noted, some acidic substances can be detected via solitary chemosensory cells and/or olfaction (e.g. electronic supplementary material, table S1), and it is possible that some species detect acids via other sensory abilities yet to be studied, such as lipophilic acidification of sensory neurons [25].

## (a) The origin of sour taste

Recently, major advances have been made in considering the evolution of sweet taste [26,27], umami taste [28,29] and bitter taste [12], as well as some of the ecological factors that drive this evolution, such as the mismatch between elemental needs of organisms (see considerations in [30]) and the foods available to those organisms [31]. These studies emphasize the many origins, losses and transitions that can occur over millions of years with regard to taste receptors as well as some of their causes. As a result, when we originally embarked upon this effort, we hypothesized that in response to the evolutionary twists and turns of vertebrate diets and needs that sour taste might have evolved more than once in vertebrates and been lost more than once as well. However, we found no examples of species unable to detect acidity in foods. And, the species in which the ability to detect acidity has been documented are phylogenetically very widespread. Together, these results are most reconcilable with a single origin and, so far, no losses of the ability to detect and respond to acidity, particularly acidity in foods.

The origin of sour taste or, more precisely, acid taste, measured here as a function of species' responses to acidity,

Proc. R. Soc. B 289: 20211918

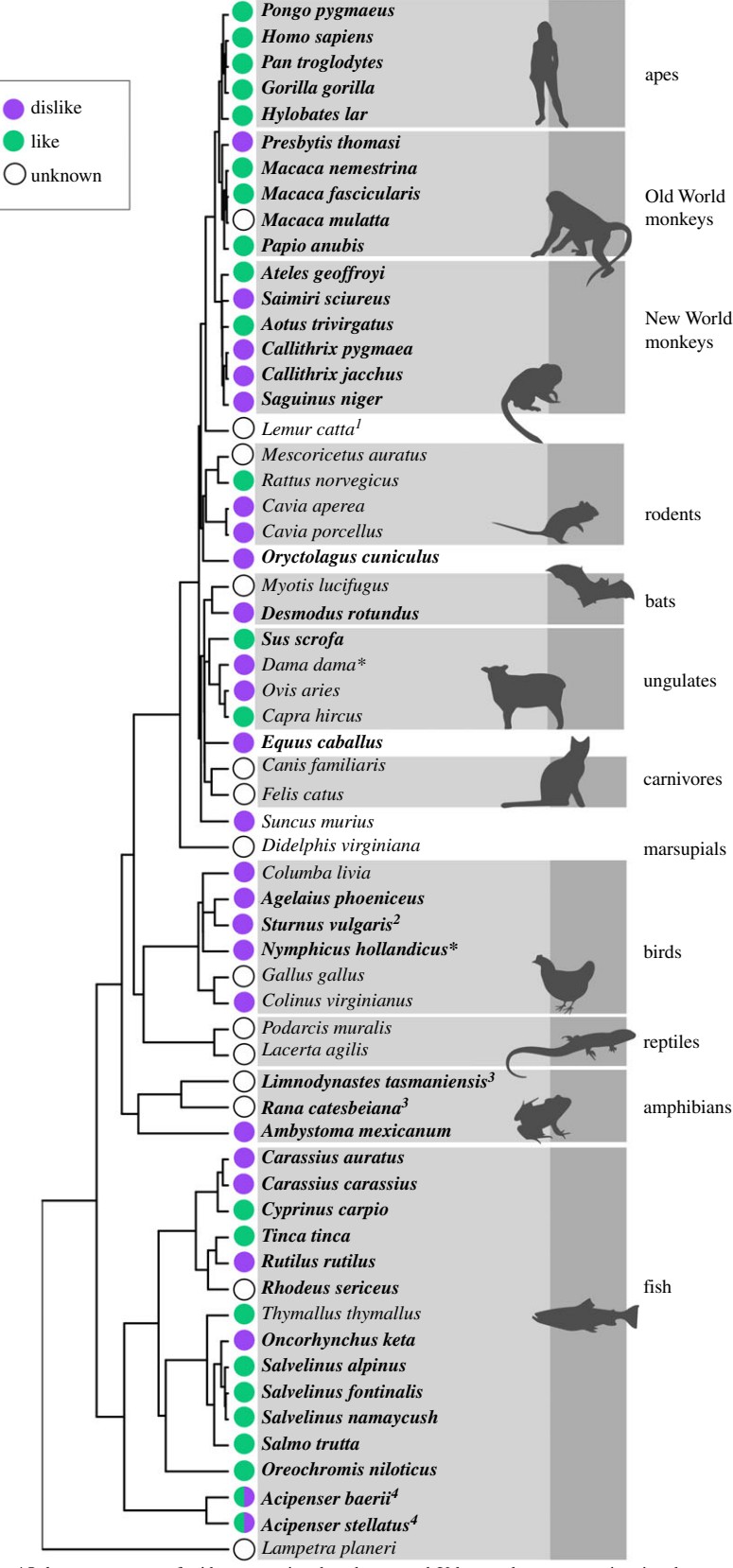

*Only a narrow range of acid concentrations have been tested. Valence at low concentrations is unknown.

1 *Lemur catta* have only been tested at very high acid concentrations.

2 *Sturnus vulgaris* males have a higher threshold of tolerance than females for acidic foods.

3 *Rana catesbeiana* and *Limnodynastes tasmaniensis* are seemingly indifferent to high concentrations of citric acid (100 nM, readily accept it, but do not show preference for it). Low concentrations of acid (< 26 mN) have not been tested and valence at these concentrations is unknown.

4 *Acipenser baerii* and *A. stellatus* show attraction to acidic foods but reject these same foods once in their mouth.

**Figure 1.** Phylogeny of vertebrate clades for which the ability to detect acid has been tested with a mapping of the valence of the reaction to acidic foods onto that phylogeny. Here 'valence' corresponds to whether or not a species likes (relative to some control) acidic foods or drinks at concentrations that are relevant to dietary preferences (see electronic supplementary material, table S2). Where possible, we focused on tests using citric acid (species names in bold). Citric acid can only be detected via sour taste, as opposed to other acids which can be detected via taste and aroma (acetic), tastes other than sour (L-aspartic) or astringency (tannic; see electronic supplementary material, table S1). (Online version in colour.)

like the origin of taste itself, is certainly ancient among vertebrates. In general, taste (gustation) is mediated by receptor cells clustered into sensory organs known as taste buds. Taste buds are found in all extant vertebrate groups, except hagfish [32,33], in which they have been secondarily lost [34]. Although solitary chemosensory cells predate them phylogenetically, taste buds evolved independently from such cells [34]. Ancestrally they were restricted to the oropharyngeal cavity [35], as they are in terrestrial vertebrates, but in some actinopterygian fishes they have secondarily spread onto external body surfaces.

The genes associated with sour taste receptors (such as OTOP1) are present in both vertebrates and invertebrates and were, therefore, almost certainly present in the first vertebrate species with taste buds. Consequently, it is possible that the earliest vertebrates already possessed the ability to detect acidic substances via sour taste receptors. Indeed, we found that a variety of modern fishes are sensitive to acidic substances, including lamprey, chondrosteans and most teleosts studied, and that the ability to taste acidic substances appears to be present throughout vertebrates (figure 1; electronic supplementary material, table S2) [36]. This suggests that acid perception was present in the common ancestor of jawless fishes and gnathostomes. In other words, the perception of acidic substances (and probably other tastes) is at least as old as vertebrates (see below). Given the phylogenetic breadth of species known to sense acidic substances (figure 1; electronic supplementary material, table S2), parsimony leads us to hypothesize a single origin and conservation of the ability to detect and respond to acidic substances, possibly through sour taste receptors, throughout vertebrate history. However, the paucity of species represented in electronic supplementary material, table S2 makes such a conclusion tentative. Until the ability to detect acid is studied in more species, we cannot exclude the possibility of losses and gains, particularly given how common losses have been for other taste receptors. For example, both sweet and umami taste receptors have both been lost repeatedly in particular vertebrate clades [37], in association with pseudogenization (preceded by reductions of the importance of those tastes to survival and reproduction). In addition, after having been lost in birds, sweet taste was regained at least once (through conversion of the umami/amino acid taste receptor into a dual-purpose umami-sweet taste receptor) [26] and potentially more than once [38]. Bitter taste receptors have been lost entirely in cetaceans [39]. In addition, the number of bitter taste receptors and the compounds to which they bind evolved, often rapidly, both among and within mammal and bird orders [13]. In this light, it seems likely that more evolutionary change has occurred in sour taste than has yet been discovered. We predict that tests for acid taste, even simple preference tests, are likely to yield surprises in the coming years.

## (b) The functional significance of acid taste in ancestral vertebrates

We can only speculate about the conditions that led ancestral vertebrates (ancestral fish), or their progenitors, to evolve acid taste; however, living fishes provide clues. Acid–base balance is a fundamental aspect of physiological homeostasis in all vertebrates because blood and tissue acidity can have severe or lethal effects on organismal function. Aquatic organisms are particularly at risk for sudden challenges to their acid–base regulation owing to extreme variation in dissolved carbon dioxide ($CO_2$) levels in both freshwater and marine habitats. High levels of $CO_2$ (hypercarbia) can occur, for example, because of poor water mixing, thick surface vegetation, high biomass, thermal stratification and microbial anaerobic metabolism [40]. Dissolved $CO_2$ forms carbonic acid (facilitated by carbonic anhydrase in tissues), which dissociates into hydrogen and bicarbonate ions and decreases water pH, which is indicative of $CO_2$ concentrations in water. Areas of hypercarbia tend to be areas of low pH [40]. Remarkably, virtually all fish have the ability to respond plastically to these dangerous conditions using two physiological mechanisms: rapid alkalosis of internal fluids with metabolically produced buffers and longer-term acclimation through the use of ion channels to generate a net efflux of $H^+$ ions. These abilities are inferred to be ancestral for all fishes [40]. Given the probable ubiquity of hypercarbia in ancestral aquatic environments and the severe consequences of acidosis, we can infer the action of strong selection on ancestral fishes for the ability to sense acidic environments. The evolution of acid-sensitive taste buds in the oropharynx, where they are well-positioned to assess local pH by monitoring the respiratory water stream, might be explained in this context.

Extending the argument further, it is likely that acid taste was the first gustatory sense to evolve. This is supported by the primary necessity of sensing acidic environments, as noted, and also by the fact that the earliest, jawless vertebrates were filter feeders that strained food particles from the water indiscriminately [41]. We hypothesize that only later, when vertebrates evolved predatory selection of particular prey types could sweet, salty and umami tastes play a role in assessing food value and palatability. Moreover, the OTOP gene family that encodes acid receptors is evolutionarily conserved, with an origin in an ancestor of extant vertebrate and invertebrate species [14,17,42].

The putative origin of taste buds within the oropharynx of ancestral fishes for acid detection would have preadapted (AKA 'exapted') them for a role in food assessment, given that food particles released during capture and processing would be immediately available (it is typically during this oral phase of feeding that food is rejected/accepted based on gustatory cues). The historical circumstances under which acid taste receptors transitioned from a purely environmental monitoring function to a role in food sensing and evaluation is hinted at by the foraging biology of the modern sea catfish, *Plotonis japonicus*. These fish use external taste buds on barbels to locate hidden benthic prey by sensing patches of acidic water generated by the prey's respiratory $CO_2$ [43,44]. More generally, many modern fish use barbels and external taste buds to locate potential food items and to determine if they are worth ingesting, although once brought into their mouths, they are often rejected, including the case of food pellets flavoured with citric acid [36]. As such, external taste appears to be less discriminatory in fish than is intra-oral taste.

## (c) The persistence of acid taste in terrestrial vertebrates

Our explanation for the origin of the ability to detect acidic foods does not explain why such an ability persisted once vertebrates adapted to life on land. One possible explanation for

the role of detection and, most frequently, aversion to acidic foods in extant terrestrial vertebrate species is that it prevents these animals from ingesting highly acidic foods such as fruits [8,45]. But this hypothesis has no fewer than three problems. First, dangerously acidic foods are not very common in nature (unripe fruits tend to be described as 'sour' by humans but are more likely to be astringent or bitter) [46]. Second, in many cases species respond aversively to only modestly acidic foods that, unless ingested in extraordinary quantities, are unlikely to be sufficiently damaging to the animal so as to be a strong selective pressure. Third, the damages that could accrue from eating acidic foods should accrue in those species that prefer such foods. One potential harm of eating acidic foods might be damage to the teeth. Dentists know this in a modern context, but such damage has also been observed in the fossil record. For example, some individuals of *Homo habilis* in Olduvai Gorge show evidence of tooth wear in line with expectations from damage associated with acidic foods [47]. To date, there does not, however, seem to be evidence that such damage is associated with increased mortality (and hence the potential for selection), whether in hominins or other taxa. Yet, before we dismiss the 'dangerous acid' hypothesis (and we should note that the authors on this paper differ in their perspectives on its explanatory potential) there is at least one set of circumstances for which acidity might, even at low concentrations, potentially be dangerous.

For animals that are foregut fermenters, acidity in food could alter the composition of the gut microbiome and alter digestion. More specifically, acidity in the foregut could favour *Lactobacillus* and *Acetobacter* species that are acid-tolerant and produce lactic acid and acetic acid, respectively (and hence further lower the pH of the habitats in which they thrive [11,48]). Species of these genera are known for their ability to hinder the growth of other taxa (for this reason, they are used in food preservation). It has been hypothesized that this ability could stall fermentation by the microbes on which foregut fermenters rely for digestion [49]. Therefore, we hypothesize that foregut fermenters are more likely to detect and avoid acidity in their foods compared to hindgut fermenters. Many of the species for which we have data and that are able to detect acidity and dislike acidic foods do indeed tend to be foregut fermenters [50]. For example, foregut-fermenting *Presbytis thomasi* is reported to prefer fruits with higher pH compared to three other sympatric hindgut-fermenting primate species [51]. However, a more systematic examination of foregut and hindgut fermenters is necessary to robustly test this hypothesis.

## (d) Why acidic foods became attractive to some species

We hypothesize that relatively acidic foods became pleasing for species (the preference–aversion function shifted right) when consumption of acidic foods was adaptive. Why acidic foods were adaptive is likely to have differed from one case to another. We consider hypotheses (and tests of such hypotheses) in more detail for four different cases.

Our **case 1** is that of night monkeys (*Aotus trivirgatus*). In 1977, Glaser hypothesized that the preference of these monkeys for concentrated acids, including acetic acid, might relate to their foraging ecology and diets [52]. Night monkeys eat fruits, and essentially all ripe fruits are fermented to some degree [53,54]. Short-chain fatty acids, such as acetic acid, are a universal by-product of rot/fermentation, and lactic acid

bacteria additionally produce lactic acid as a by-product of their metabolism. It may be the case, Glaser hypothesizes [52] that at night fruits that are more heavily fermented, and thus have higher concentrations of acid by-products, are easier to smell (this hypothesis seems testable, at least in a zoo context, but appears to have gone untested). If right, night monkeys with a preference for such fruits might be more likely to survive.

Our **case 2** is that of pigs, which can detect acidity in their food [55] and, at least in the contexts so far studied, tend to prefer it [56,57]. In nature, diverse species of suids including the wild relatives of domestic pigs, forage for hidden items on, within, and below the ground level detritus using olfaction as a distal sense to find foods (the pig leads with its nose). Strong olfactory cues are associated with some foods that are not acidic (e.g. roots, tubers, truffles [58]). But strong olfactory cues are also produced by food that has begun to rot or ferment, such as fallen fruits and carrion. Wild pigs have a strong attraction to fermented baits, especially fermented or soured corn and in some places wild pigs rely heavily on carrion [59]. Whereas plant and animal remains that are rotted by many kinds of microbes can contain metabolites that negatively affect health [60], those rotted by lactic acid bacteria or acetic acid bacteria are less likely to pose problems because lactic acid and acetic acid kill many potentially harmful microbes [61]. Hence, a pig that was pleased by acidic, rotten foods might, therefore, have access to more safe foods of which to avail itself. In the case of both night monkeys and pigs, it would be useful to test just which microbes are most active in the rotten foods they eat and how and if pigs choose among rotten foods differing in their acidity (and hence the abundance of acid-producing bacteria).

Our **case 3** is that of the subset of diurnal monkeys and apes, scattered across the primate phylogenetic tree, that have shown a preference for slightly to highly acidic foods. The common ancestor of monkeys and apes lost the ability to produce vitamin C, roughly 61–74 Ma [62], after divergence from strepsirrhines (several of which appear to show aversion to acids, or at least to high concentrations of citric acid [63]). Most mammals produce their own vitamin C [64]. This loss is due to pseudogenization of the GLO gene (L-gulono-γ-lactone oxidase), necessary to produce vitamin C [65]. It has been frequently hypothesized that this pseudogenization occurred because the common ancestor of monkeys and apes consumed enough vitamin C in its fruit-heavy diet so as to no longer need to produce its own vitamin C [62,66,67]. Breslin [10] added to this account the novel hypothesis that once a subset of primate species shifted to diets that included less vitamin C, they were then at a disadvantage (a modern human manifestation of this disadvantage can be seen in the history of scurvy, the disease caused by the lack of vitamin C; [68]). This might have been particularly likely to be true in omnivorous species living in habitats with fewer fruit-bearing trees (such as grasslands), but it would also be true in habitats in which only a subset of fruits reliably contains vitamin C. In such contexts, any individuals in populations of species that were more likely to prefer acidic foods might have increased their probability of encountering vitamin C which in itself has no taste other than the sourness that marks its acidity (vitamin C is ascorbic acid). Vitamin C is not in all acidic foods, but all foods that are high in vitamin C are acidic, including a subset of fruits. Breslin's hypothesis is germane to a number of primate species that forage in

open habitats. It could account, for instance, for the evolution of sour taste preferences in olive baboons (*Papio anubis*; [69]). It is also germane to the story of the common ancestor of chimpanzees, gorillas and modern humans, a species that lived roughly 10 Ma, a story that we now consider in more detail.

In our **case 4**, we zoom in on hominids. Both chimpanzees and humans either instinctively prefer acidic foods or readily learn to prefer them (electronic supplementary material, table S2). Many of the fruits eaten by chimpanzees are either sweet and sour or just sour [70], as subjectively judged by Toshisada Nishida (using his own tongue) in studying chimpanzees in the Mahale population, as well as by researchers studying other populations (e.g. [71,72]). Unfortunately, no studies appear to have compared the frequency of sour fruits in nature to those in chimpanzee diets, which would be a useful step. Gorillas also consume acidic foods. A long-term study by Sabater-Pi of lowland gorillas at the Rio Muni site in Equatorial Guinea found that many of the fruits preferred by the gorillas were acidic. The main fruit consumed by the gorillas was a species of the genus *Aframomum*. The fruits of some species of *Aframomum* taste sweet [70], but the species that Sabater-Pi tasted at Rio Muni tasted (to Sabater-Pi) sweet and very acidic [72]. In addition, at least one extinct species of *Homo*, *Homo habilis*, appears to have consumed acidic foods [47]. As for the more distant relatives of humans, orangutans and gibbons, the most comprehensive study to date suggests that they eat and, in this case, seem to prefer acidic fruits. In 1993, Ungar [51] compared the fruit choices of orangutans, gibbons, macaques and presbytis at a single site. Ungar found that the orangutans, gibbons and macaques were all more likely to choose acidic fruit (pH less than 4.5) than were the presbytis monkeys (see above). The fruits that were most likely to be eaten by orangutans and gibbons were the most acidic ones. In the future, it would be useful to study the taste preferences of extant ape species in zoo environments by presenting them with the same fruit differing in sweetness and acidity.

Until more studies are conducted on taste preferences in apes, we must consider the evolution of acid taste preference in hominids cautiously (and without more studies of other African primates, we cannot fully rule out the possibility that acid preference evolved in the ancestor of African monkeys and apes and then was lost in some species, such as the monkey *P. thomasi*). To date, the observed patterns are in line with what might be expected if the most recent common ancestor (MRCA) of humans, chimpanzees, gorillas, orangutans and gibbons evolved a preference for acidic foods. If this is the case, any explanation for the origin of the preference for acidic fruits must relate to the biology of a common ancestor living a primarily arboreal lifestyle. However, if the preference for acid tastes evolved later (a possibility we cannot yet preclude), in the MRCA of chimpanzees, gorillas and humans (subsequent to the divergence of its lineage from that of orangutans and gibbons), it would be tempting to link the origin of a preference to the spread of open habitats during a time of shrinking forests. During this time, it is thought that this common ancestor (i) shifted to a lifestyle that involved more time spent on the ground and (ii) relied more heavily on non-forest habitats with lower densities of fruit trees and higher densities of plants with stronger physical and chemical defenses such as fibre and toxins [73–77]. As this occurred, a preference for acidic foods could have offered two potential advantages. It would have rewarded individuals for finding fruits that were likely to have vitamin C (see above). It would have also encouraged them to consume fruits on the ground that tend to be in later stages of rot [53], which tends to be called fermentation when it yields preferred outcomes (though the distinction is fuzzy).

## (e) Consequences of sour taste preferences for hominins

Regardless of whether rotting fruits played a role in the shift of the acid preference curve in hominins, we hypothesize that the existence of acid taste preference may have strongly influenced the later relationship between hominins and rotten fruits and other rotten foods. Based on studies in the laboratory, three groups of microorganisms compete during the rot of fruits [78], single-celled budding yeasts (most of which are from the Saccharomycetales clade of fungi), filamentous fungi (such as *Penicillium*) and lactic acid bacteria. While all of these organisms produce short-chain fatty acids when they ferment fruit, yeasts also tend to produce alcohol, and lactic acid bacteria produce lactic acid. Rotten fruits that become dominated by filamentous fungi can be dangerous [79]. However, rotten fruits that become dominated by yeasts and lactic bacteria are often 'improved' from the perspective of consumers. Rot due to lactic acid bacteria and yeasts often increases food caloric, free amino acid and vitamin content and hence improves digestibility by breaking down fibre and plant toxins [80–84]. Therefore, in challenging nutritional environments, fruits rotted by yeasts or lactic acid bacteria likely represented a valuable food source that could increase chances of survival [4]. If the acid-preference of the MRCA (whenever acquired) allowed it to more readily consume heavily fermented fruit, or at least the subset of that fruit rotted by lactic acid bacteria, they might have been able to take advantage of a novel source of safe calories.

There exists molecular evidence that the last common ancestor of gorillas, chimpanzees and humans consumed fermented fruits. For example, a single amino acid replacement in the *ADH4* gene in the lineage shared by humans and African apes resulted in a 40-fold improvement in ethanol oxidation [85]. This change would have allowed the MRCA to consume yeast-fermented fruits on the ground with higher concentrations of both ethanol and acids [85] without concomitant neurological toxicity (or drunkenness; [53]). This ability may have allowed the MRCA to survive and reproduce more effectively in nutritionally challenging, seasonal environments, particularly as climate change resulted in more fragmented and open habitats. At about the same time, the MRCA acquired a third copy of the HCA3 gene encoding G protein-coupled receptors for hydroxycarboxylic acids, such as lactic acid, produced by the fermentation of dietary carbohydrates by lactic acid bacteria [86]. While this gene is found in all great apes, it is most strongly activated in chimpanzees, gorillas and humans, with humans exhibiting the strongest effects, suggesting that, in some form acid-producing bacteria (and the detection of their products) played a larger role in apes than in other primates and in humans than in non-human apes. As has been considered elsewhere, a fondness for acidic foods, particularly when combined with preferences for umami tastes, may have predisposed ancestral humans to eventual intentional control of rotting to yield more favourable outcomes, which is to say, fermentation [4,87].

## (f) Broadening back out

The great hope is that, now that some aspects of the taste receptors for sour taste are beginning to be understood, the genetics and hence molecular evolution of sour can be studied and compared across large numbers of species. We are hopeful. But we would also be remiss if we did not mention that this has been the hope before. Sour taste is challenging. What is clear is that sour taste is ancient and, at least so far as we have yet detected, rarely if ever lost among vertebrates. However, the preference–aversion function for acidic compounds appears to have shifted both among vertebrates in general and within primates in particular. Just how such shifts occur is not understood, but in the context of the story of human evolution the shift that led to our own human preference–aversion functions for acidic foods appears likely to have had great consequences for the human relationship to fermented foods and drinks. It is possible that ancient human preference–aversion functions for acidic foods evolved so as to make our ancestors more likely to be able to appreciate certain ripe or rotting fruits that contained acids that inhibit harmful microbes or even fruits that have been intentionally fermented. But it is also possible that our preference–aversion functions simply shifted to guide us to vitamin C and was, thus, preadapted for ancient humans to love fermenting foods.

Data accessibility. This article has no additional data.

Authors' contributions. H.E.R.F.: conceptualization, visualization, writing—original draft, writing—review and editing; K.A.: conceptualization, methodology, project administration, writing—original draft, writing—review and editing; M.T.: data curation, formal analysis, visualization, writing—original draft, writing—review and editing; P.M.: investigation, methodology, writing—original draft, writing—review and editing; E.R.L.: conceptualization, writing—original draft, writing—review and editing; L.M.N.: conceptualization, data curation, formal analysis, methodology, project administration, validation, visualization, writing—original draft, writing—review and editing; K.S.: conceptualization, data curation, investigation, methodology, writing—original draft, writing—review and editing; P.A.S.B.: conceptualization, methodology, validation, writing—original draft, writing—review and editing; R.R.D.: conceptualization, data curation, formal analysis, investigation, methodology, project administration, supervision, validation, visualization, writing—original draft, writing—review and editing.

All authors gave final approval for publication and agreed to be held accountable for the work performed therein.

Competing interests. We declare we have no competing interests.

Funding. We received no funding for this study.

Acknowledgements. We thank CIFAR's 'Humans and the Microbiome' programme for research support. Thanks also to Michael Kaspari who helped spur our consideration of the evolution of taste and its consequences. Thanks also to four very helpful anonymous reviewers; you made this a much better paper.

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
