## [Peer Review File · Proceedings of the Royal Society B: Biological Sciences]

Review History

RSPB-2021-1918.R0 (Original submission)

Review form: Reviewer 1

Recommendation

Accept with minor revision (please list in comments)

Scientific importance: Is the manuscript an original and important contribution to its field?

Excellent

General interest: Is the paper of sufficient general interest?

Excellent

Quality of the paper: Is the overall quality of the paper suitable?

Good

Is the length of the paper justified?

Yes

Should the paper be seen by a specialist statistical reviewer?

No

Do you have any concerns about statistical analyses in this paper? If so, please specify them explicitly in your report.

No

It is a condition of publication that authors make their supporting data, code and materials available - either as supplementary material or hosted in an external repository. Please rate, if applicable, the supporting data on the following criteria.

Is it accessible?

Yes

Is it clear?

Yes

Is it adequate?

Yes

Do you have any ethical concerns with this paper?

No

Comments to the Author

In this manuscript, Frank and colleagues provide one a much needed review exploring the evolution of sour taste in vertebrates, with a particular focus on humans and non-human primates. This topic has not been well explored in previous studies of sensory function and ecology, despite its clear importance. The authors compile a database of vertebrates for which the ability to detect acidity in food has been tested and code these taxa on a phylogeny to offer a preliminary investigation of the evolution of sour taste across vertebrates. The team of authors bridges numerous relevant disciplines and is well-qualified to explore this topic.

Overall, I think this is a good and necessary review that will stimulate future research in sensory evolution. I was super excited to read it! The first half of the manuscript is very strong and tightly written – it does an excellent job summarizing what is known about sour taste and the challenges in studying it. I also particularly enjoyed the phylogeny in Figure 1 and the discussions of the “Origin of Sour Taste” and the “Functional Significance of Acid Taste in Ancestral Vertebrates.”

The second half of the manuscript is weaker and a little less organized. It is a lot more narrative in format, which is not a format I am used to with Proceedings B articles. I think these subsequent sections (“Why acidic foods became attractive”, “Consequences of sour Taste Preferences for Hominins”, etc.) should provide explicit, testable hypotheses to avoid being “just so” stories. I know they mention some future directions (e.g., lines 365-368), but I think making testable hypotheses a more conscious focus of the manuscript will really help stimulate research in this topic.

I have 2 other major comments and then some minor points:

- Figure 1: The figure legend (like, dislike, unknown/variable) is confusing, and the legend states that it is grouping the “three models of sour taste evolution”. Nowhere else in the manuscript do you mention “three models of sour taste evolution”. The three options also differ from how you coded vertebrate clades (“sour tasters”, “sour non-tasters”, “data deficient”). I understand this coding system is different from what you are trying to depict in the figure, but you should use the legend to better explain what this figure is depicting.

- What is the status of Pongo? There is a lack of consistency in the manuscript regarding Pongo and sour taste. In Figure 1 and Table S2, you describe Pongo as detecting sour taste and disliking it. For that info, you cite [65], which is Remis (2002). Remis (2002) doesn't mention Pongo. Is that the correct citation?

Also, however, in lines 310-312, you state that “In contrast, there is no clear evidence that orangutans or gibbons and siamangs prefer acidic foods (P. Fan, C. Knott, E. Vogel, S. Wich, personal communication)”. But your figure/table stated that Pongo disliked acidic food. Can you please clarify?

- Line 238: “we dismiss the ‘dangerous acid,’ hypothesis” - I don’t think you need the comma after acid.

- Line 255 (and 404): Please use a consistent formatting (e.g., for headers, capitalizing, colon vs. em dash).

- Please be sure to italicize gene names (e.g., ADH4 in line 340).

- Line 347-351: What gene?

- Line 378: I think you’re missing “been” - “may have BEEN faster”

- Lines 370-386: The use of author-focused statements so many times throughout this paragraph is disruptive (e.g., “Breslin pointed out”, “While Amato et al. [4], point out”, “Amato et al. [4] argue”, “Breslin [10] points out”. I understand that both of the lead authors of those citations are authors of the current paper, but it creates a weird back and forth to have it occurring so frequently in a single paragraph. Can this paragraph be rephrased to focus on the findings rather than the authors?

Review form: Reviewer 2

Recommendation

Accept with minor revision (please list in comments)

Scientific importance: Is the manuscript an original and important contribution to its field?

Good

General interest: Is the paper of sufficient general interest?

Good

Quality of the paper: Is the overall quality of the paper suitable?

Good

Is the length of the paper justified?

Yes

Should the paper be seen by a specialist statistical reviewer?

No

Do you have any concerns about statistical analyses in this paper? If so, please specify them explicitly in your report.

No

It is a condition of publication that authors make their supporting data, code and materials available - either as supplementary material or hosted in an external repository. Please rate, if applicable, the supporting data on the following criteria.

Is it accessible?

Yes

Is it clear?

Yes

Is it adequate?

Yes

Do you have any ethical concerns with this paper?

No

Comments to the Author

The authors have investigated the origin of sour taste through a literature review and discussed about the origin of sour taste and the evolution of a preference for acidic foods. Although sour taste is generally considered as an aversive taste, I agree with the idea that many animals including humans can enjoy the foods with good acidity and sour taste is not only an aversive but also a preferable taste. The work provides new significant insights to understand the evolution of sour taste. I have a few comments, which I outline below.

Figure 1.

As the authors say, the sour taste of acids can be both pleasing and displeasing depending on their concentrations. Thus, we cannot conclude whether an animal dislikes acids without doing the preference tests in a wide range of acid concentrations. For example, the authors say that a horse dislikes acids on Fig. 1, but the paper cited (ref 91) used only one acid (acetic acid) at one concentration (0.16 mL / 100 mL). Thus, the possibilities that a horse prefers other acids or acetic acid at the lower concentrations still remain.

Table S1.

I don't think that "SCFA" means a taste. I know that fatty acids can activate GPR40 and GPR120 on the tongue of mice, but don't know if this signal means a taste.

Line 142 on Supplemental Materials.

No example was given after "e.g.".

Review form: Reviewer 3

Recommendation

Accept with minor revision (please list in comments)

Scientific importance: Is the manuscript an original and important contribution to its field?

Good

General interest: Is the paper of sufficient general interest?

Acceptable

Quality of the paper: Is the overall quality of the paper suitable?

Good

Is the length of the paper justified?

Yes

Should the paper be seen by a specialist statistical reviewer?

No

Do you have any concerns about statistical analyses in this paper? If so, please specify them explicitly in your report.

No

It is a condition of publication that authors make their supporting data, code and materials available - either as supplementary material or hosted in an external repository. Please rate, if applicable, the supporting data on the following criteria.

Is it accessible?

Yes

Is it clear?

Yes

Is it adequate?

Yes

Do you have any ethical concerns with this paper?

No

Comments to the Author

In the manuscript “The Evolution of Sour Taste” the authors provide an overview of the state of the literature on what we know about sour taste in animals, as well as a collection of the existing evidence about which animals like or dislike sour flavors, which the authors place into phylogenetic context and discuss in an evolutionary framework. The review provides a valuable overview of a taste that has received little attention, especially compared to bitter taste. The authors’ writing is very accessible (perhaps occasionally bordering on colloquial) and I enjoyed reading the manuscript. I expect that it will be of interest to a fairly broad readership, despite its focus on primates.

In order to broaden the review’s appeal, I’d like to make the following suggestions:

I would have liked to see a lot more citations throughout the review – it wasn’t always clear to me where information was coming from, so parts of the manuscript read a bit like just-so stories. A few examples follow, but I encourage the authors to be more liberal with citing throughout the manuscript.

For example, in line 300, the authors imply that hamadryas baboons have a sour taste preference, but don’t provide a citation for this and hamadryas baboons do not appear in Figure 1 (unless *Papio anubis* is meant to be *Papio hamadryas*?). The reader has to go to the supplemental table to find the reference for this, which is a bit cumbersome.

Another statement was in lines 335-337: “Given the choice of these kinds of fermented fruits, modern apes appear to choose those that are lactic or alcoholic, but not those that have been fermented by filamentous fungi” – no citation is provided here, so it’s unclear where the evidence for it can be found.

The section on human consumption of carrion also needs additional citations, for example, it would be great to direct readers to references in which its consumption is “hotly contested” (line 390) and to where it is hypothesized that sour taste is a signal for absence of *Clostridia* (line 395).

L107-108 – The authors state that their second step, coding vertebrate classes into tasters and non-tasters, could be simplified, (“In practice, as will be revealed in the results, realities of the data allowed this step to be simplified.”), however, I didn’t follow how this was simplified and I didn’t see any further explanations of this in the results (maybe because, this being a review paper, there was no discrete results section). It becomes a bit clearer when reading the supplemental text, however, without a note in the main text, the reader doesn’t know to look for this additional information in the supplements (instead one looks for “results”). I realize that

the space for the main text is probably limited, but I would have liked to see this explained properly here, instead of having a vague reference (“will be revealed”), to avoid confusion for the reader. If this is not possible, instead of referring to “results” it’s probably more appropriate to refer to the supplemental text.

Line 242 onward – I was pleased to see a discussion of foregut fermenters, this came to mind for me when first reading the article abstract. However, this section contains not a single citation, even though there has been work documenting this. As this is a review article, readers would benefit from being directed to the existing research here and I suspect many readers will read the manuscript for this reason.

L 383 – While fermentation of roots and tubers to improve their flavor may be important, it should be noted that these items are often cooked by humans today. In the presence of controlled fire (which humans have had for some time), cooking a tuber would be much faster than waiting for it to ferment. Is there evidence that these items are routinely fermented, rather than cooked today?

L 420 – The authors conclude on a suggestion that ancient humans may have been pre-adapted to love “rotten foods,” however, earlier in the article the authors seem to distinguish quite carefully between “fermented” and “rotten,” suggesting that sour taste may have helped human distinguish between the two and avoid rotten food. It seems to counter this argument to end on a statement about humans having been pre-adapted to love “rotten” (rather than “fermented”) foods.

Minor comments:

Line 136-137 – “And, the species in which the ability to detect acidity are phylogenetically very widespread” – is there a word or phrase missing in this sentence? “has been documented”, maybe?

Line 204 – Should it read “with an origin in the ancestor of...” here?

Lines 339 & 340 – ADH4 gene name should be italicized.

Line 378 – “may have been faster”?

Suppl. L142 – there is an “e.g.,” but no examples are listed in the parentheses

Table S2 – common names are provided for some species but not for all (entries for bats and birds, for example, do not include common names). For consistency, the authors should include common names for all species.

Review form: Reviewer 4

Recommendation

Accept with minor revision (please list in comments)

Scientific importance: Is the manuscript an original and important contribution to its field?

Good

General interest: Is the paper of sufficient general interest?

Good

Quality of the paper: Is the overall quality of the paper suitable?

Good

Is the length of the paper justified?

Yes

Should the paper be seen by a specialist statistical reviewer?

No

Do you have any concerns about statistical analyses in this paper? If so, please specify them explicitly in your report.

No

It is a condition of publication that authors make their supporting data, code and materials available - either as supplementary material or hosted in an external repository. Please rate, if applicable, the supporting data on the following criteria.

Is it accessible?

Yes

Is it clear?

Yes

Is it adequate?

Yes

Do you have any ethical concerns with this paper?

No

Comments to the Author

Comments are attached as two-page pdf file (see Appendix A).

Decision letter (RSPB-2021-1918.R0)

08-Oct-2021

Dear Dr Dunn:

Your manuscript has now been peer reviewed and the reviews have been assessed by an Associate Editor. The reviewers' comments (not including confidential comments to the Editor) and the comments from the Associate Editor are included at the end of this email for your reference. As you will see, the reviewers and the Editors have raised some concerns with your manuscript and we would like to invite you to revise your manuscript to address them.

Research ethics:

Use of animals and field studies:

It is a condition of publication that you make available the data and research materials supporting the results in the article (<https://royalsociety.org/journals/authors/author-guidelines/#data>). Datasets should be deposited in an appropriate publicly available repository and details of the associated accession number, link or DOI to the datasets must be included in the Data Accessibility section of the article (<https://royalsociety.org/journals/ethics-policies/data-sharing-mining/>). Reference(s) to datasets should also be included in the reference list of the article with DOIs (where available).

Please submit a copy of your revised paper within three weeks. If we do not hear from you within this time your manuscript will be rejected. If you are unable to meet this deadline please let us know as soon as possible, as we may be able to grant a short extension.

Best wishes,
Editor, Proceedings B
mailto:proceedingsb@royalsociety.org

Associate Editor, Dr Amanda Melin
Board Member: 1
Comments to Author(s):
(There are no comments.)

The authors provide an interesting and comprehensive review of the evolution of sour taste, and I am excited for the potential of this contribution to the field of sensory ecology.

This manuscript has now been reviewed by four experts, who provide many useful suggestions. I share their sentiments of enthusiasm for the content, ideas, and accessibility. I also agree with the constructive points raised, and ask that the authors carefully address them in a revised version. In particular, I agree that the sections on primate preferences in natural environments are overstated. In addition, multiple reviewers raise the need for increased precision in the discussion of existing literature. Overall, there is a need, in several areas, to make clearer what is speculation (or could be new testable hypotheses), and what claims are evidence-based, with additional references to primary literature. Please also see requests for revision to Fig 1.

I look forward to reading a revised version of this exciting manuscript.

Reviewer(s)' Comments to Author:
Referee: 1

Comments to the Author(s)

In this manuscript, Frank and colleagues provide one a much needed review exploring the evolution of sour taste in vertebrates, with a particular focus on humans and non-human primates. This topic has not been well explored in previous studies of sensory function and ecology, despite its clear importance. The authors compile a database of vertebrates for which the ability to detect acidity in food has been tested and code these taxa on a phylogeny to offer a preliminary investigation of the evolution of sour taste across vertebrates. The team of authors bridges numerous relevant disciplines and is well-qualified to explore this topic.

Overall, I think this is a good and necessary review that will stimulate future research in sensory evolution. I was super excited to read it! The first half of the manuscript is very strong and tightly written – it does an excellent job summarizing what is known about sour taste and the challenges in studying it. I also particularly enjoyed the phylogeny in Figure 1 and the discussions of the “Origin of Sour Taste” and the “Functional Significance of Acid Taste in Ancestral Vertebrates.”

The second half of the manuscript is weaker and a little less organized. It is a lot more narrative in format, which is not a format I am used to with Proceedings B articles. I think these subsequent

sections (“Why acidic foods became attractive”, “Consequences of sour Taste Preferences for Hominins”, etc.) should provide explicit, testable hypotheses to avoid being “just so” stories. I know they mention some future directions (e.g., lines 365-368), but I think making testable hypotheses a more conscious focus of the manuscript will really help stimulate research in this topic.

I have 2 other major comments and then some minor points:

- Figure 1: The figure legend (like, dislike, unknown/variable) is confusing, and the legend states that it is grouping the “three models of sour taste evolution”. Nowhere else in the manuscript do you mention “three models of sour taste evolution”. The three options also differ from how you coded vertebrate clades (“sour tasters”, “sour non-tasters”, “data deficient”). I understand this coding system is different from what you are trying to depict in the figure, but you should use the legend to better explain what this figure is depicting.

- What is the status of Pongo? There is a lack of consistency in the manuscript regarding Pongo and sour taste. In Figure 1 and Table S2, you describe Pongo as detecting sour taste and disliking it. For that info, you cite [65], which is Remis (2002). Remis (2002) doesn't mention Pongo. Is that the correct citation?

Also, however, in lines 310-312, you state that “In contrast, there is no clear evidence that orangutans or gibbons and siamangs prefer acidic foods (P. Fan, C. Knott, E. Vogel, S. Wich, personal communication)”. But your figure/table stated that Pongo disliked acidic food. Can you please clarify?

- Line 238: “we dismiss the ‘dangerous acid,’ hypothesis” – I don’t think you need the comma after acid.

- Line 255 (and 404): Please use a consistent formatting (e.g., for headers, capitalizing, colon vs. em dash).

- Please be sure to italicize gene names (e.g., ADH4 in line 340).

- Line 347-351: What gene?

- Line 378: I think you’re missing “been” – “may have BEEN faster”

- Lines 370-386: The use of author-focused statements so many times throughout this paragraph is disruptive (e.g., “Breslin pointed out”, “While Amato et al. [4], point out”, “Amato et al. [4] argue”, “Breslin [10] points out”. I understand that both of the lead authors of those citations are authors of the current paper, but it creates a weird back and forth to have it occurring so frequently in a single paragraph. Can this paragraph be rephrased to focus on the findings rather than the authors?

Referee: 2

Comments to the Author(s)

The authors have investigated the origin of sour taste through a literature review and discussed about the origin of sour taste and the evolution of a preference for acidic foods. Although sour taste is generally considered as an aversive taste, I agree with the idea that many animals including humans can enjoy the foods with good acidity and sour taste is not only an aversive but also a preferable taste. The work provides new significant insights to understand the evolution of sour taste. I have a few comments, which I outline below.

Figure 1.

As the authors say, the sour taste of acids can be both pleasing and displeasing depending on their concentrations. Thus, we cannot conclude whether an animal dislikes acids without doing the preference tests in a wide range of acid concentrations. For example, the authors say that a

horse dislikes acids on Fig. 1, but the paper cited (ref 91) used only one acid (acetic acid) at one concentration (0.16 mL / 100 mL). Thus, the possibilities that a horse prefers other acids or acetic acid at the lower concentrations still remain.

Table S1.

I don't think that "SCFA" means a taste. I know that fatty acids can activate GPR40 and GPR120 on the tongue of mice, but don't know if this signal means a taste.

Line 142 on Supplemental Materials.

No example was given after "e.g.".

Referee: 3

Comments to the Author(s)

In the manuscript "The Evolution of Sour Taste" the authors provide an overview of the state of the literature on what we know about sour taste in animals, as well as a collection of the existing evidence about which animals like or dislike sour flavors, which the authors place into phylogenetic context and discuss in an evolutionary framework. The review provides a valuable overview of a taste that has received little attention, especially compared to bitter taste. The authors' writing is very accessible (perhaps occasionally bordering on colloquial) and I enjoyed reading the manuscript. I expect that it will be of interest to a fairly broad readership, despite its focus on primates.

In order to broaden the review's appeal, I'd like to make the following suggestions:

I would have liked to see a lot more citations throughout the review – it wasn't always clear to me where information was coming from, so parts of the manuscript read a bit like just-so stories. A few examples follow, but I encourage the authors to be more liberal with citing throughout the manuscript.

For example, in line 300, the authors imply that hamadryas baboons have a sour taste preference, but don't provide a citation for this and hamadryas baboons do not appear in Figure 1 (unless *Papio anubis* is meant to be *Papio hamadryas*?). The reader has to go to the supplemental table to find the reference for this, which is a bit cumbersome.

Another statement was in lines 335-337: "Given the choice of these kinds of fermented fruits, modern apes appear to choose those that are lactic or alcoholic, but not those that have been fermented by filamentous fungi" – no citation is provided here, so it's unclear where the evidence for it can be found.

The section on human consumption of carrion also needs additional citations, for example, it would be great to direct readers to references in which its consumption is "hotly contested" (line 390) and to where it is hypothesized that sour taste is a signal for absence of *Clostridia* (line 395).

L107-108 – The authors state that their second step, coding vertebrate classes into tasters and non-tasters, could be simplified, ("In practice, as will be revealed in the results, realities of the data allowed this step to be simplified."), however, I didn't follow how this was simplified and I didn't see any further explanations of this in the results (maybe because, this being a review paper, there was no discrete results section). It becomes a bit clearer when reading the supplemental text, however, without a note in the main text, the reader doesn't know to look for this additional information in the supplements (instead one looks for "results"). I realize that the space for the main text is probably limited, but I would have liked to see this explained properly here, instead of having a vague reference ("will be revealed"), to avoid confusion for the reader. If this is not possible, instead of referring to "results" it's probably more appropriate to refer to the supplemental text.

Line 242 onward – I was pleased to see a discussion of foregut fermenters, this came to mind for me when first reading the article abstract. However, this section contains not a single citation, even though there has been work documenting this. As this is a review article, readers would

benefit from being directed to the existing research here and I suspect many readers will read the manuscript for this reason.

L 383 – While fermentation of roots and tubers to improve their flavor may be important, it should be noted that these items are often cooked by humans today. In the presence of controlled fire (which humans have had for some time), cooking a tuber would be much faster than waiting for it to ferment. Is there evidence that these items are routinely fermented, rather than cooked today?

L 420 – The authors conclude on a suggestion that ancient humans may have been pre-adapted to love “rotten foods,” however, earlier in the article the authors seem to distinguish quite carefully between “fermented” and “rotten,” suggesting that sour taste may have helped human distinguish between the two and avoid rotten food. It seems to counter this argument to end on a statement about humans having been pre-adapted to love “rotten” (rather than “fermented”) foods.

Minor comments:

Line 136-137 – “And, the species in which the ability to detect acidity are phylogenetically very widespread” – is there a word or phrase missing in this sentence? “has been documented”, maybe?

Line 204 – Should it read “with an origin in the ancestor of...” here?

Lines 339 & 340 – ADH4 gene name should be italicized.

Line 378 – “may have been faster”?

Suppl. L142 – there is an “e.g.,” but no examples are listed in the parentheses

Table S2 – common names are provided for some species but not for all (entries for bats and birds, for example, do not include common names). For consistency, the authors should include common names for all species.

Referee: 4

Comments to the Author(s)

Comments are attached as two-page pdf file.

Author's Response to Decision Letter for (RSPB-2021-1918.R0)

See Appendix B.

RSPB-2021-1918.R1 (Revision)

Review form: Reviewer 1

Recommendation

Accept as is

Scientific importance: Is the manuscript an original and important contribution to its field?

Excellent

General interest: Is the paper of sufficient general interest?

Excellent

Quality of the paper: Is the overall quality of the paper suitable?

Good

Is the length of the paper justified?

Yes

Should the paper be seen by a specialist statistical reviewer?

No

Do you have any concerns about statistical analyses in this paper? If so, please specify them explicitly in your report.

No

It is a condition of publication that authors make their supporting data, code and materials available - either as supplementary material or hosted in an external repository. Please rate, if applicable, the supporting data on the following criteria.

Is it accessible?

Yes

Is it clear?

Yes

Is it adequate?

Yes

Do you have any ethical concerns with this paper?

No

Comments to the Author

In this manuscript, the authors give an excellent review of what is currently known regarding the prevalence of sour taste across vertebrates, preferences regarding sour taste, and offer several interesting scenarios and hypotheses for variation in sour taste in different taxa. I reviewed a previous version of this manuscript, and Frank and colleagues did an admirable job addressing my comments and the comments of the other reviewers. I think they have revised their manuscript into a compelling article with a number of testable hypotheses that will be of great interest across ecology and evolution.

Review form: Reviewer 4

Recommendation

Accept with minor revision (please list in comments)

Scientific importance: Is the manuscript an original and important contribution to its field?

Good

General interest: Is the paper of sufficient general interest?

Good

Quality of the paper: Is the overall quality of the paper suitable?

Good

Is the length of the paper justified?

Yes

Should the paper be seen by a specialist statistical reviewer?

No

Do you have any concerns about statistical analyses in this paper? If so, please specify them explicitly in your report.

No

It is a condition of publication that authors make their supporting data, code and materials available - either as supplementary material or hosted in an external repository. Please rate, if applicable, the supporting data on the following criteria.

Is it accessible?

Yes

Is it clear?

Yes

Is it adequate?

Yes

Do you have any ethical concerns with this paper?

No

Comments to the Author

The authors have satisfied my comments/concerns, and I recommend acceptance. But first, I have a few small comments:

1. The word "of" is duplicated on line 4 of paragraph 2 of p. 5
2. Convention is to capitalize the word "Gorge" of Olduvai Gorge (p. 8)
3. The correct spelling is *Presbytis thomasi* (p. 8)
4. Colobines, like every vertebrate on Earth, including ruminants, are "monogastric", so I am confused by the distinction. If you want to distinguish between large, sacculated stomachs and simple stomachs, then fine, but let's not classify chambers in a ruminant's stomach with multiple stomachs (p. 8)
5. Better to specify lowland gorillas to avoid any confusion with mountain gorillas (p. 10)

Decision letter (RSPB-2021-1918.R1)

23-Dec-2021

Dear Dr Dunn

I am pleased to inform you that your revised Review manuscript RSPB-2021-1918.R1 entitled "The Evolution of Sour Taste" has been accepted for publication in Proceedings B.

The referees do not recommend any further revision apart from some very minor changes. Therefore, please make the changes, proof-read your manuscript carefully and upload your final files for publication. Because the schedule for publication is very tight, it is a condition of publication that you submit the revised version of your manuscript within 14 days. If you do not think you will be able to meet this date please let me know immediately.

To upload your manuscript, log into <http://mc.manuscriptcentral.com/prsb> and enter your Author Centre, where you will find your manuscript title listed under "Manuscripts with Decisions." Under "Actions," click on "Create a Revision." Your manuscript number has been appended to denote a revision.

You will be unable to make your revisions on the originally submitted version of the manuscript. Instead, upload a new version through your Author Centre.

1) A text file of the manuscript (doc, txt, rtf or tex), including the references, tables (including captions) and figure captions. Please remove any tracked changes from the text before submission. PDF files are not an accepted format for the "Main Document".

2) A separate electronic file of each figure (tiff, EPS or print-quality PDF preferred). The format should be produced directly from original creation package, or original software format. Please note that PowerPoint files are not accepted.

3) Electronic supplementary material: this should be contained in a separate file from the main text and the file name should contain the author's name and journal name, e.g. `authorname_procb_ESM_figures.pdf`

All supplementary materials accompanying an accepted article will be treated as in their final form. They will be published alongside the paper on the journal website and posted on the online figshare repository. Files on figshare will be made available approximately one week before the accompanying article so that the supplementary material can be attributed a unique DOI. Please see: <https://royalsociety.org/journals/authors/author-guidelines/>

4) Data-Sharing and data citation

It is a condition of publication that data supporting your paper are made available. Data should be made available either in the electronic supplementary material or through an appropriate repository. Details of how to access data should be included in your paper. Please see <https://royalsociety.org/journals/ethics-policies/data-sharing-mining/> for more details.

<http://datadryad.org/submit?journalID=RSPB&manu=RSPB-2021-1918.R1> which will take you to your unique entry in the Dryad repository.

Once again, thank you for submitting your manuscript to Proceedings B and I look forward to receiving your final version. If you have any questions at all, please do not hesitate to get in touch.

Best wishes for now and 2022,
Innes Cuthill

Professor Innes Cuthill
Editor, Proceedings B
mailto:proceedingsb@royalsociety.org

Associate Editor

Comments to Author:

The revised manuscript is much improved and satisfactorily addresses all of the referees' previous concerns. (But please see very minor editorial notes from one reviewer). I am excited to see this manuscript in print and congratulate the authors on an exciting and timely review.

Reviewer(s)' Comments to Author:

Referee: 1

Comments to the Author(s)

In this manuscript, the authors give an excellent review of what is currently known regarding the prevalence of sour taste across vertebrates, preferences regarding sour taste, and offer several interesting scenarios and hypotheses for variation in sour taste in different taxa. I reviewed a previous version of this manuscript, and Frank and colleagues did an admirable job addressing my comments and the comments of the other reviewers. I think they have revised their manuscript into a compelling article with a number of testable hypotheses that will be of great interest across ecology and evolution.

Referee: 4

Comments to the Author(s)

The authors have satisfied my comments/concerns, and I recommend acceptance. But first, I have a few small comments:

1. The word "of" is duplicated on line 4 of paragraph 2 of p. 5
2. Convention is to capitalize the word "Gorge" of Olduvai Gorge (p. 8)
3. The correct spelling is *Presbytis thomasi* (p. 8)
4. Colobines, like every vertebrate on Earth, including ruminants, are "monogastric", so I am confused by the distinction. If you want to distinguish between large, sacculated stomachs and simple stomachs, then fine, but let's not classify chambers in a ruminant's stomach with multiple stomachs (p. 8)
5. Better to specify lowland gorillas to avoid any confusion with mountain gorillas (p. 10)

Decision letter (RSPB-2021-1918.R2)

05-Jan-2022

Dear Dr Dunn

I am pleased to inform you that your manuscript entitled "The Evolution of Sour Taste" has been accepted for publication in Proceedings B.

Data Accessibility section

Open Access

Paper charges

Sincerely,

Proceedings B

Appendix A

Review of RSPB-2021-1918 titled, “The Evolution of Sour Taste” by Frank et al.

General comments.

I accepted this review assignment with great interest, and I was not disappointed. The scope and writing style are commendable, and I appreciate how the manuscript alternates between what we know and what we don't know. The best reviews lack backward to propel a field forward, and this paper does as much. I have no comments of substance other than to gently challenge a few claims and matters of grammar. But I will say this:

1. Oral biologists are likely to howl in protest if this paper is published without mentioning the costs to our teeth. Acidic soft drinks are a major cause of tooth enamel wear for many human populations, and there is even evidence of significant acid etching on the teeth of some hominins, notably *Homo habilis* (Puech, 1984). So, yes, we like sour foods but it comes at a cost, though probably not enough to affect fitness.

Specific comments.

1. Lines 304-320. Here the authors make broad claims about primate preferences in natural settings, and it is bridge too far in my view. It is sufficient to say that systematic data are rare, but the subjective impressions of human observers affirm that some edible foods are sour. I would steer the authors to a paper by Nishida et al. (2000), which (a) contradicts the notion that chimpanzees prefer sour foods, although % annual diet is problematic as measure of preference; (b) shows that plant tissues aside from fruits can taste sour; and (c) reports a very different taste sensation for *Aframomum*, a relatively speciose genus [note the correct spelling, *Afra*-, not *Afro*-]. I would also note from their table that figs don't taste sour, which raises questions about potential incompatibilities between acidity and their unique reproductive biology. The authors also overlook semi-quantitative data published by Ungar (1995), which directly contradicts their claim that “there is no clear evidence that orangutans or gibbons and siamangs prefer acidic foods” (lines 311-312). In Ungar's data set, gibbons and orangutans appear to prefer fruits with a pH < 4.25 compared to those with a higher pH. And bolstering Ungar's data set from Sumatra are data published by Lucas and Corlett (1991), who showed that *Garcinia* fruits (a strong favorite of gibbons, orangutans, and humans) can have a lemon-like pH of 2.5-3.0. None of this undermines the overarching point being made by the authors, but it does affect the factual accuracy and comprehensiveness of their review.

Last, the authors should take note of Ungar's data for the colobine monkey *Presbytis*, which supports their claim that colobines should avoid acidic fruits.

2. Lines 370-372: I had a strong negative reaction to these words, which imply willful fermentation by an ape living 10 million years ago. The authors really must draw a sharp distinction between an ape eating naturally fermented foods and a human being capable of directing fermentation, even if Epipalaeolithic humans did it passively without an understanding of microbial life. And on Line 373: I don't think Breslin was referring to any hominid other than *Homo sapiens*, but he is a co-author on this manuscript so I defer

to him. Still, I find these lines weirdly vague and arm-waving when there is zero evidence that any antecedent of *Homo sapiens* fermented fruit with the intent of doing so. And again, on Line 377, I really can't believe that anyone would seriously claim that a nonhuman primate has "the ability to ferment foods". These are jaw-dropping words that impute an astonishing and unsubstantiated level of cognition for nonhuman primates.

3. Line 383: The pH of African tubers and other plant underground storage organs is 6.7 +/- 0.4, so yes, they are decidedly more neutral than leaves (5.6 +/- 0.7) or fruits (5.5 +/- 1.0) in the diets of Kibale primates (Dominy and Lucas 2004). This same paper found that smaller, redder fruits tend to be more acidic—many tropical ecologists would describe small red fruits as "bird-adapted fruits."

And if memory serves, Janzen (1977) argued that acidic fruits were those with prolonged development trajectories. Such fruits are more vulnerable to fungal pathogens and rotting before achieving ripeness. It could be useful here for the authors to address ideas for why some tropical fruits in primate diets are more substantially acidic than others (cf. data in Lucas and Corlett 1991; Ungar 1995).

4. Line 430: author name should be written "Katz SE"

Works cited.

Dominy, N. J., and P. W. Lucas. 2004. Significance of color, calories, and climate to the visual ecology of catarrhines. *American Journal of Primatology* 62:189-207.

Janzen, D. H. 1977. Why fruits rot, seeds mold, and meat spoils. *American Naturalist* 111:691-713.

Lucas PW, Corlett RT. 1991. Quantitative aspects of the relationship between dentitions and diets. In: Vincent JFV, Lillford PJ, editors. *Feeding and the texture of food*. Cambridge: Cambridge University Press. p 93–121.

Nishida, T. et al. 2000. Tastes of chimpanzee plant foods. *Curr Anthropol* 41:431-438.

Puech, P.-F. 1984. Acidic-food choice in *Homo habilis* at Olduvai. *Curr Anthropol* 25:349-350.

Ungar PS. 1995. Fruit preferences of four sympatric primate species at Ketambe, northern Sumatra, Indonesia. *Int J Primatol* 16:221–245.

Appendix B

R1: In this manuscript, Frank and colleagues provide one a much needed review exploring the evolution of sour taste in vertebrates, with a particular focus on humans and non-human primates. This topic has not been well explored in previous studies of sensory function and ecology, despite its clear importance. The authors compile a database of vertebrates for which the ability to detect acidity in food has been tested and code these taxa on a phylogeny to offer a preliminary investigation of the evolution of sour taste across vertebrates. The team of authors bridges numerous relevant disciplines and is well-qualified to explore this topic. Overall, I think this is a good and necessary review that will stimulate future research in sensory evolution. I was super excited to read it! The first half of the manuscript is very strong and tightly written – it does an excellent job summarizing what is known about sour taste and the challenges in studying it. I also particularly enjoyed the phylogeny in Figure 1 and the discussions of the “Origin of Sour Taste” and the “Functional Significance of Acid Taste in Ancestral Vertebrates.”

Authors: Thank you.

R1: The second half of the manuscript is weaker and a little less organized. It is a lot more narrative in format, which is not a format I am used to with Proceedings B articles. I think these subsequent sections (“Why acidic foods became attractive”, “Consequences of sour Taste Preferences for Hominins”, etc.) should provide explicit, testable hypotheses to avoid being “just so” stories. I know they mention some future directions (e.g., lines 365-368), but I think making testable hypotheses a more conscious focus of the manuscript will really help stimulate research in this topic.

Authors: This is a great point. We have now made the organization of this section more explicit (case 1, case 2, etc...) and shorter (we reduced its length by three paragraphs). We have also made sure to include testable hypotheses for each of these sections. We are grateful for this reviewer comment. We think these changes greatly strengthened the paper.

R1: I have 2 other major comments and then some minor points:

- Figure 1: The figure legend (like, dislike, unknown/variable) is confusing, and the legend states that it is grouping the “three models of sour taste evolution”. Nowhere else in the manuscript do you mention “three models of sour taste evolution”. The three options also differ from how you coded vertebrate clades (“sour tasters”, “sour non-tasters”, “data deficient”). I understand this coding system is different from what you are trying to depict in the figure, but you should use the legend to better explain what this figure is depicting.

Authors: Thank you. We have reworked Figure 1 and the legend for Figure 1. We redefined the valence assignments in Table S2 and updated the legend for Figure 1 to more explicitly describe what we mean by the terms. We have also added a more extensive description of our methods in the supplement. The legend now reads: “Phylogeny of vertebrate clades, calling attention to those featured here, with a mapping of the valence of the reaction to sour foods onto that phylogeny. Here “valence” corresponds to whether or not a species likes (relative to some control) acidic foods or drinks at concentrations that are relevant to dietary preferences (see Supplemental Methods and Table S2)”

R1: What is the status of Pongo? There is a lack of consistency in the manuscript regarding Pongo and sour taste. In Figure 1 and Table S2, you describe Pongo as detecting sour taste and disliking it. For that info, you cite [65], which is Remis (2002). Remis (2002) doesn't mention Pongo. Is that the correct citation?

Authors: We have corrected this citation to Ungar (1995) and include a note in the table indicating that the prevailing data for *Pongo* is based on the pH of foods in their diet rather than explicit acid preference testing.

R1: Also, however, in lines 310-312, you state that “In contrast, there is no clear evidence that orangutans or gibbons and siamangs prefer acidic foods (P. Fan, C. Knott, E. Vogel, S. Wich, personal communication)”. But your figure/table stated that Pongo disliked acidic food. Can you please clarify?

Authors: We apologize for these inconsistencies. The Remis paper does not include orangutans and our personal communications with orangutan researchers did not provide any clear consensus. However, we now include new references that more quantitatively suggest that orangutans do have a preference for sour foods. We have updated the text and figure to reflect this new information.

R1: - Line 238: “we dismiss the ‘dangerous acid,’ hypothesis” – I don’t think you need the comma after acid.

Author: Change made.

R1: - Line 255 (and 404): Please use a consistent formatting (e.g., for headers, capitalizing, colon vs. em dash).

Author: Change made

R1: - Please be sure to italicize gene names (e.g., ADH4 in line 340).

Author: Change made

R1: - Line 347-351: What gene?

Author: This is the HCA3 gene. We have now included the name in the text.

R1: - Line 378: I think you’re missing “been” – “may have BEEN faster”

Author: Oops! Thank you.

R1: - Lines 370-386: The use of author-focused statements so many times throughout this paragraph is disruptive (e.g., “Breslin pointed out”, “While Amato et al. [4], point out”, “Amato et al. [4] argue”, “Breslin [10] points out”. I understand that both of the lead authors of those citations are authors of the current paper, but it creates a weird back and forth to have it occurring so frequently in a single paragraph. Can this paragraph be rephrased to focus on the findings rather than the authors?

Authors: We have streamlined this text so as to be less herky jerky. We removed all of the references to the authors names and used a more standard citation/referencing approach. Thank you.

R2: The authors have investigated the origin of sour taste through a literature review and discussed about the origin of sour taste and the evolution of a preference for acidic foods. Although sour taste is generally considered as an aversive taste, I agree with the idea that many animals including humans can enjoy the foods with good acidity and sour taste is not only an aversive but also a preferable taste. The work provides new significant insights to understand the evolution of sour taste. I have a few comments, which I outline below.

Author: Thank you.

R2: Figure 1.

As the authors say, the sour taste of acids can be both pleasing and displeasing depending on their concentrations. Thus, we cannot conclude whether an animal dislikes acids without doing the preference tests in a wide range of acid concentrations. For example, the authors say that a horse dislikes acids on Fig. 1, but the paper cited (ref 91) used only one acid (acetic acid) at one concentration (0.16 mL / 100 mL). Thus, the possibilities that a horse prefers other acids or acetic acid at the lower concentrations still remain.

Authors: Thank you. We have reworked Figure 1 and the legend for Figure 1. We redefined the valence assignments to reflect the range of acids tested. We now include in the range of concentrations tested in Table 2 and explicitly indicate when the valence assignments were based on only a narrow range of acid concentrations vs a wide range of acid concentrations (including lower concentrations). We added a paragraph in the methods section describing how we assigned valence.

R2: Table S1. I don't think that “SCFA” means a taste. I know that fatty acids can activate GPR40 and GPR120 on the tongue of mice, but don't know if this signal means a taste.

Authors: Acetic acid stimulates SCFA receptors on taste cells but at this time it is unclear if there is a separate SCFA taste apart from sourness. To be on the safe side we include SCFA (given the potential that it might be tasted). But we don't feel strongly and are glad to omit it if the reviewer thinks it is warranted.

R2: Line 142 on Supplemental Materials. No example was given after “e.g.”.

Authors: We removed this sentence entirely, since we no longer included the referenced acids.

R3: In the manuscript “The Evolution of Sour Taste” the authors provide an overview of the state of the literature on what we know about sour taste in animals, as well as a collection of the existing evidence about which animals like or dislike sour flavors, which the authors place into phylogenetic context and discuss in an evolutionary framework. The review provides a valuable overview of a taste that has received little attention, especially compared to bitter taste. The authors’ writing is very accessible (perhaps occasionally bordering on colloquial) and I enjoyed reading the manuscript. I expect that it will be of interest to a fairly broad readership, despite its focus on primates.

Author: Thank you.

R3: In order to broaden the review’s appeal, I’d like to make the following suggestions:

Author: Thank you for this very productive framing.

R3: I would have liked to see a lot more citations throughout the review – it wasn’t always clear to me where information was coming from, so parts of the manuscript read a bit like just-so stories.

Author: Thank you. We have 1) reduced the length of the most speculative final part of the paper, 2) considerably increased the number of citations in that section and 3) made sure that each of our hypotheses is associated with one or more readily tested hypotheses.

R3: A few examples follow, but I encourage the authors to be more liberal with citing throughout the manuscript.

Authors: Thank you. We have taken this guidance to heart and increased the number of citations throughout.

R3: For example, in line 300, the authors imply that hamadryas baboons have a sour taste preference, but don’t provide a citation for this and hamadryas baboons do not appear in Figure 1 (unless *Papio anubis* is meant to be *Papio hamadryas*?). The reader has to go to the supplemental table to find the reference for this, which is a bit cumbersome.

Authors: Thank you for pointing this out. We have corrected the text to indicate that we meant the subspecies typically called olive baboons (which was *Papio hamadryas anubis* and is now *Papio anubis*), not *Papio hamadryas*. The citation is included in the supplementary table, but we now also include it in the text.

R3: Another statement was in lines 335-337: “Given the choice of these kinds of fermented fruits, modern apes appear to choose those that are lactic or alcoholic, but not those that have been fermented by filamentous fungi” – no citation is provided here, so it’s unclear where the evidence for it can be found.

Authors: Thank you. We have now added a citation to this statement.

R3: The section on human consumption of carrion also needs additional citations, for example, it would be great to direct readers to references in which its consumption is “hotly contested” (line 390) and to where it is hypothesized that sour taste is a signal for absence of Clostridia (line 395).

Authors: We have now removed this section.

R3: L107-108 – The authors state that their second step, coding vertebrate classes into tasters and non-tasters, could be simplified, (“In practice, as will be revealed in the results, realities of the data allowed this step to be simplified.”), however, I didn’t follow how this was simplified and I didn’t see any further explanations of this in the results (maybe because, this being a review paper, there was no discrete results section). It becomes a bit clearer when reading the supplemental text, however, without a note in the main text, the reader doesn’t know to look for this additional information in the supplements (instead one looks for “results”). I realize that the space for the main text is probably limited, but I would have liked to see this explained properly here, instead of having a vague reference (“will be revealed”), to avoid confusion for the reader. If this is not possible, instead of referring to “results” it’s probably more appropriate to refer to the supplemental text.

Authors: We have reworded this section to be less cryptic. We briefly and explicitly explain our methods and point readers to the Supplementary methods for more details.

R3: Line 242 onward – I was pleased to see a discussion of foregut fermenters, this came to mind for me when first reading the article abstract. However, this section contains not a single citation, even though there has been work documenting this. As this is a review article, readers would benefit from being directed to the existing research here and I suspect many readers will read the manuscript for this reason.

Authors: Thank you. We now cite Ungar 1995, Ginane et al. 2011, Calsamiglia et al. 2012, and Overend et al. 2016

R3: L 383 – While fermentation of roots and tubers to improve their flavor may be important, it should be noted that these items are often cooked by humans today. In the presence of controlled fire (which humans have had for some time), cooking a tuber would be much faster than waiting for it to ferment. Is there evidence that these items are routinely fermented, rather than cooked today?

Authors: Per other comments, we have removed this section of the paper.

R3: L 420 – The authors conclude on a suggestion that ancient humans may have been pre-adapted to love “rotten foods,” however, earlier in the article the authors seem to distinguish quite carefully between “fermented” and “rotten,” suggesting that sour taste may have helped human distinguish between the two and avoid rotten food. It seems to counter this argument to end on a statement about humans having been pre-adapted to love “rotten” (rather than “fermented”) foods.

Authors: The distinction between rotting and fermentation is messy. In general, fermentation is just rotting that yields an outcome that the consumer likes. We’ve now added the following sentence, which we hope is clarifying...

“It would have also encouraged them to consume fruits on the ground that tend to be in later stages of rot [52], which tends to be called fermentation when it yields preferred outcomes (though the distinction is fuzzy).”

R3: Line 136-137 – “And, the species in which the ability to detect acidity are phylogenetically very widespread” – is there a word or phrase missing in this sentence? “has been documented”, maybe?

Authors: Thank you. We fixed this sentence.

R3: Line 204 – Should it read “with an origin in the ancestor of...” here?

Authors: Thank you. Fixed.

R3: Lines 339 & 340 – ADH4 gene name should be italicized.

Authors: Thank you. Fixed.

R3: Line 378 – “may have been faster”?

Authors: Thank you for catching this.

Suppl. L142 – there is an “e.g.,” but no examples are listed in the parentheses

Authors: We removed this sentence entirely, since we no longer included the referenced acids.

R3: Table S2 – common names are provided for some species but not for all (entries for bats and birds, for example, do not include common names). For consistency, the authors should include common names for all species.

Authors: Thank you, we have added all common names, where appropriate.

R4: Review of RSPB-2021-1918 titled, “The Evolution of Sour Taste” by Frank et al.
General comments.

I accepted this review assignment with great interest, and I was not disappointed. The scope and writing style are commendable, and I appreciate how the manuscript alternates between what we know and what we don't know. The best reviews lack backward to propel a field forward, and this paper does as much. I have no comments of substance other than to gently challenge a few claims and matters of grammar.

Author: Thank you for these kind thoughts.

R4: But I will say this:

1. Oral biologists are likely to howl in protest if this paper is published without mentioning the costs to our teeth. Acidic soft drinks are a major cause of tooth enamel wear for many human populations, and there is even evidence of significant acid etching on the teeth of some hominins, notably *Homo habilis* (Puech, 1984). So, yes, we like sour foods but it comes at a cost, though probably not enough to affect fitness.

Author: This is a good point. We've now added new sentences “First, acid can damage teeth. Dentists know this in a modern context, but such damage has also been observed in the fossil record. For example, some individuals of *Homo habilis* in Olduvai gorge show evidence of tooth wear in line with expectations from damage associated with acidic foods. At least in theory, such wear might precipitate tooth infections and death.”

R4: Lines 304-320. Here the authors make broad claims about primate preferences in natural settings, and it is bridge too far in my view. It is sufficient to say that systematic data are rare, but the subjective impressions of human observers affirm that some edible foods are sour.

RRD: We have shortened this section and better cited it. Thank you for this feedback.

R4: I would steer the authors to a paper by Nishida et al. (2000), which (a) contradicts the notion that chimpanzees prefer sour foods

Authors: We now cite Nishida et al. (2000), and reference this section much better in general, but disagree slightly with the reviewers statement. Nishida found that many of the fruits consumed by chimpanzees were sweet and sour or, more rarely, just sour. In one year (1995), such fruits made up roughly a quarter of the diet of chimpanzees. Nishida does not actually comment on this finding (he shows it as a figure, pasted below) but does go on to indicate that “Izawa and Itani (1966) stated that two of the four favorite fruits of the chimpanzees of Kasakati, Tanzania, were sweet-sour” and then also to note similar findings for another field site.

FIG. 1. The "taste world" of wild chimpanzees; time spent eating food items classified in terms of taste. Left, 1994; right, 1995.

R4: although % annual diet is problematic as measure of preference;

Authors: We now state.... "Unfortunately, no studies appear to have compared the frequency of sour fruits in nature to those in chimpanzee diets (in order to truly document preference as opposed to tolerance), which would be a useful step.

R4...(continuing) [Nishida] shows that plant tissues aside from fruits can taste sour

Authors: Than you. This is a useful observation.

R4.... [Nishida] reports a very different taste sensation for Aframomum, a relatively speciose genus [note the correct spelling, Afra-, not Afro-].

Authors: We now note that different species of Aframomum may have different tastes. Thank you for the spelling fix.

R4: I would also note from their table that figs don't taste sour, which raises questions about potential incompatibilities between acidity and their unique reproductive biology.

Authors: That is very interesting. We aren't sure it is worth bring up in our manuscript, but it has caused us to explore figs in more detail.

R4: The authors also overlook semi-quantitative data

published by Ungar (1995), which directly contradicts their claim that “there is no clear evidence that orangutans or gibbons and siamangs prefer acidic foods” (lines 311-312). In Ungar’s data set, gibbons and orangutans appear to prefer fruits with a pH < 4.25 compared to those with a higher pH. And bolstering Ungar’s data set from Sumatra are data published by Lucas and Corlett (1991), who showed that *Garcinia* fruits (a strong favorite of gibbons, orangutans, and humans) can have a lemon-like pH of 2.5-3.0. None of this undermines the overarching point being made by the authors, but it does affect the factual accuracy and comprehensiveness of their review.

Last, the authors should take note of Ungar’s data for the colobine monkey *Presbytis*, which supports their claim that colobines should avoid acidic fruits.

Authors: We thank the reviewers for bringing this literature to our attention. We have now included it and used it to reframe our discussion of sour taste, apes and African monkeys.

R4: 2. Lines 370-372: I had a strong negative reaction to these words, which imply willful fermentation by an ape living 10 million years ago. The authors really must draw a sharp distinction between an ape eating naturally fermented foods and a human being capable of directing fermentation, even if Epipalaeolithic humans did it passively without an understanding of microbial life. And on Line 373: I don’t think Breslin was referring to any hominid other than *Homo sapiens*, but he is a co-author on this manuscript so I defer

to him. Still, I find these lines weirdly vague and arm-waving when there is zero evidence that any antecedent of *Homo sapiens* fermented fruit with the intent of doing so. And again, on Line 377, I really can’t believe that anyone would seriously claim that a nonhuman primate has “the ability to ferment foods”. These are jaw-dropping words that impute an astonishing and unsubstantiated level of cognition for nonhuman primates.

Authors: We have now shortened the discussion of controlled fermentation to two sentences that point readers to other published studies and make very modest claims. We write, simply....

“As has been considered elsewhere, a fondness for acidic foods, particularly when combined with preferences for umami tastes, may have predisposed ancestral humans to eventual control of fermentation.”

R4: 3. Line 383: The pH of African tubers and other plant underground storage organs is 6.7 +/- 0.4, so yes, they are decidedly more neutral than leaves (5.6 +/- 0.7) or fruits (5.5 +/- 1.0) in the diets of Kibale primates (Dominy and Lucas 2004). This same paper found that smaller, redder fruits tend to be more acidic—many tropical ecologists would describe small red fruits as “bird-adapted fruits.”

Authors: Thank you. This is super interesting, but given the shortening of the end of the paper we decided not to mention it in this paper.

R4: And if memory serves, Janzen (1977) argued that acidic fruits were those with prolonged development trajectories. Such fruits are more vulnerable to fungal pathogens and rotting before achieving ripeness. It could be useful here for the authors to address ideas for why some tropical fruits in primate diets are more substantially acidic than others (cf. data in Lucas and Corlett 1991; Ungar 1995) .

Authors: This is a fascinating question. We have reduced the length of the section about Fermentation (per other reviewers' suggestions) but agree that this is a really interesting question.

R4: 4. Line 430: author name should be written "Katz SE"

Authors: We have corrected this citation.

R4: Works cited.

Dominy, N. J., and P. W. Lucas. 2004. Significance of color, calories, and climate to the visual ecology of catarrhines. *American Journal of Primatology* 62:189-207.

Janzen, D. H. 1977. Why fruits rot, seeds mold, and meat spoils. *American Naturalist* 111:691-713.

Lucas PW, Corlett RT. 1991. Quantitative aspects of the relationship between dentitions and diets. In: Vincent JFV, Lillford PJ, editors. *Feeding and the texture of food*. Cambridge: Cambridge University Press. p 93–121.

Nishida, T. et al. 2000. Tastes of chimpanzee plant foods. *Curr Anthropol* 41:431-438.

Puech, P.-F. 1984. Acidic-food choice in *Homo habilis* at Olduvai. *Curr Anthropol* 25:349-350.

Ungar PS. 1995. Fruit preferences of four sympatric primate species at Ketambe, northern Sumatra, Indonesia. *Int J Primatol* 16:221–245.

Authors: We have added most of these references.